# Influence of Alfvén Ion–Cyclotron Waves on the Anisotropy of Solar Wind Turbulence at Ion Kinetic Scales

Xin Wang [1,2,*] , Linzhi Huang [1], Yuxin Wang [1] and Haochen Yuan [1]

1  School of Space and Environment, Beihang University, Beijing 100083, China
2  Key Laboratory of Space Environment Monitoring and Information Processing, Ministry of Industry and Information Technology, Beijing 100804, China
*  Correspondence: wangxinpku0209@gmail.com

**Abstract:** The power spectra of the magnetic field at ion kinetic scales have been found to be significantly influenced by Alfvén ion–cyclotron (AIC) waves. Here, we study whether and how this influence of the AIC wave depends on the $\theta_{VB}$ angle (the angle between the local mean magnetic field and the solar wind velocity direction). The wavelet technique is applied to the high time-resolution (11 vectors per second) magnetic field data from *WIND* spacecraft measurements in a fast solar wind stream associated with an outward magnetic sector. It is found that around the ion kinetic scales (0.52 Hz–1.21 Hz), the power spectrum in the parallel angular bin $0° < \theta_{VB} < 10°$ has a slope of $-4.80 \pm 0.15$. When we remove the left-handed polarized AIC waves (with normalized reduced magnetic helicity smaller than $-0.9$) from the fluctuations, the spectral index becomes $-4.09 \pm 0.11$. However, the power spectrum in the perpendicular angular bin $80° < \theta_{VB} < 90°$ changes very little during the wave-removal process, and its slope remains $-3.22 \pm 0.07$. These results indicate that the influence of the AIC waves on the magnetic spectral index at the ion kinetic scales is indeed dependent on $\theta_{VB}$, which is due to the anisotropic distribution of the waves. Apparently, when the waves are removed from the original data, the spectral anisotropy weakens. This result may help us to better understand the physical nature of the spectral anisotropy around the ion scales.

**Keywords:** solar wind; turbulence; waves

## 1. Introduction

Solar wind turbulence has been studied extensively for several decades [1–3]. It is well known that in the solar wind, the magnetic fluctuations have a power-law spectrum with the Kolmogorov spectral index of $k^{-5/3}$ in the inertial range [4–7]. Around the ion kinetic scales, the spectrum is still a power-law, but the index becomes much steeper in the dissipation range [3,8–10]. The spectral index of the magnetic fluctuations in this domain reveals large variation, which ranges from $-4.5$ to $-1.5$ [11–13]. The reason for this steepening and the large variation is still subject to debate. In the Earth's magnetosphere, the magnetic spectrum has been observed to have a similar profile as in the solar wind e.g., [14–21].

The power spectrum of the magnetic field fluctuations around the ion kinetic scales has been found to be anisotropic with respect to the angle $\theta_{VB}$ (the angle between the local mean magnetic field and solar wind velocity direction). Leamon et al. [11] showed that the spectra at different $\theta_{VB}$ angles have different power levels and different spectral indices. We estimated the indices over the range 0.5–2 Hz manually in Figure 6 of Leamon et al. [11], and found that for the events with $\theta_{VB} = 23°$ and $87°$, their spectral indices are $-4.3$ and $-2.5$, respectively. Chen et al. [22] used a multi-spacecraft analysis technique to calculate the second-order structure functions at different angles to the local mean magnetic field. They found that the spectral indices of the transverse components are $-2.6$ and $-3$ at large and small angles, respectively. This spectral steepening towards small angles was considered to be in broad agreement with a critically balanced cascade of whistler or kinetic

Alfvén waves. The cascade theory predicts $k^{-7/3}$ for the perpendicular spectrum and $k^{-5}$ for the parallel spectrum [23,24]. However, it should be noted that the steepest spectral index which the structure function technique permits us to measure is $-3$ [25,26], and so it is necessary to use another method (e.g., Morlet wavelet) in order to handle a spectrum as steep as $k^{-5}$ in the parallel direction.

Alfvén ion–cyclotron (AIC) waves are often found in the solar wind. Some possible signatures of the AIC waves in the solar wind were reported many years ago [27,28]. Recently, using the high time-resolution magnetic field data from the *STEREO* mission, Jian et al. [29,30] observed strong narrow-band AIC waves in the solar wind near 1 AU. Wicks et al. [31] identified a proton–cyclotron wave storm from the WIND spacecraft observations. The normalized reduced magnetic helicity ($\sigma_m$) was used to help find AIC waves in the solar wind. He et al. [32] presented a possible signature of the AIC waves in the distribution of $\sigma_m$ as a function of $\theta_{VB}$. They found that fluctuations around the ion–cyclotron frequency had a dominant negative (positive) $\sigma_m$ for $\theta_{VB} < 30°$ ($\theta_{VB} > 150°$) in the solar wind outward (inward) sector. This finding is consistent with the presence of left-handed parallel propagating AIC waves among the solar wind fluctuations. In contrast, the right-hand perpendicular propagating waves with positive (negative) $\sigma_m$ for $40° < \theta_{VB} < 140°$ in the solar wind outward (inward) sector were identified as kinetic Alfvén waves or whistler waves. These observations have been corroborated by several authors e.g., [33–36].

Most recently, Lion et al. [37] studied the influence of coherent events (including AIC waves and coherent structures) on the spectral shape of the magnetic field fluctuations around the ion scales. They separated the coherent events from the data set by applying the Morlet wavelet transform method [38], in order to compare their individual spectra with the original spectrum of the whole time interval. The result showed that both the AIC waves and the coherent structures have a spectrum with a small bump around the ion–cyclotron frequency, while the spectra of the remaining non-coherent fluctuations do not exhibit any break. The authors concluded that a combination of waves and coherent structures determines the spectral shape of the magnetic field spectrum around the ion scales. Recently, Zhao et al. [39] found a negative correlation between the spectral index and $\sigma_m$ near the proton gyroradius scale. However, the influence of AIC waves on the spectral shape around the ion scales must depend on the angle $\theta_{VB}$, since the AIC wave characteristics are spatially anisotropic and depend on $\theta_{VB}$.

In this work, we will investigate the angular dependence of the influence of waves on the magnetic spectral index around the ion kinetic scales in the fast solar wind by using the Morlet wavelet technique. The paper is organized as follows. In Section 2, we introduce the data used in this work and the methods applied to remove the AIC waves from the original data. Then we compare the spectral index before and after the removal of the wave data in each $\theta_{VB}$ angular bin. In Section 3, we show our observational results. In Section 4, we summarize this work and discuss its consequences.

## 2. Data and Methods

We use high-resolution magnetic-field data (with a sampling of 11 vectors per second) obtained via the Magnetic Field Investigation [40] on board the *WIND* spacecraft. The data from a long-duration fast solar wind stream observed between January 14 and 20 in Year 2008 are used here. This fast stream was also analyzed by Wicks et al. [41] and Wang et al. [42] to show the spectral anisotropy of both magnetic field and solar wind velocity in the inertial range. During this time interval, the *WIND* spacecraft was located in the halo orbit around the Lagrange 1 point.

In the upper three panels of Figure 1, we show the time series of the three components of both magnetic field vectors (black) and the flow velocity vectors (blue) for the 7-day interval in geocentric solar ecliptic (GSE) coordinates. In Figure 1d, we also show the magnetic field strength (black) and proton number density (blue). The mean flow velocity and proton number density of the interval are 680 km s$^{-1}$ and 2.5 cm$^{-3}$, respectively, which are the typical characteristics of fast solar wind streams. During this interval, we see

that the magnetic field strength keeps steady, and the $x$ component of the magnetic field keeps pointing away from the Sun at about 1-day scale, although there are many small spikes superposed on it. In addition, the $x$ component of the flow velocity is still larger than 600 km s$^{-1}$ near the end of this interval, which means that the spacecraft was still located in the fast stream rather than a rarefaction region. According to these characteristics, we speculate that this stream could come from a single coronal hole. The ion–cyclotron frequency of this interval is $f_{ci} = eB/(2\pi m) = 0.06$ Hz. And the angle between the global mean magnetic field and the solar wind velocity direction is about 40°, which is close to the local Parker angle at Lagrange 1 point (∼45°). This indicates that this interval belongs to an outward magnetic sector.

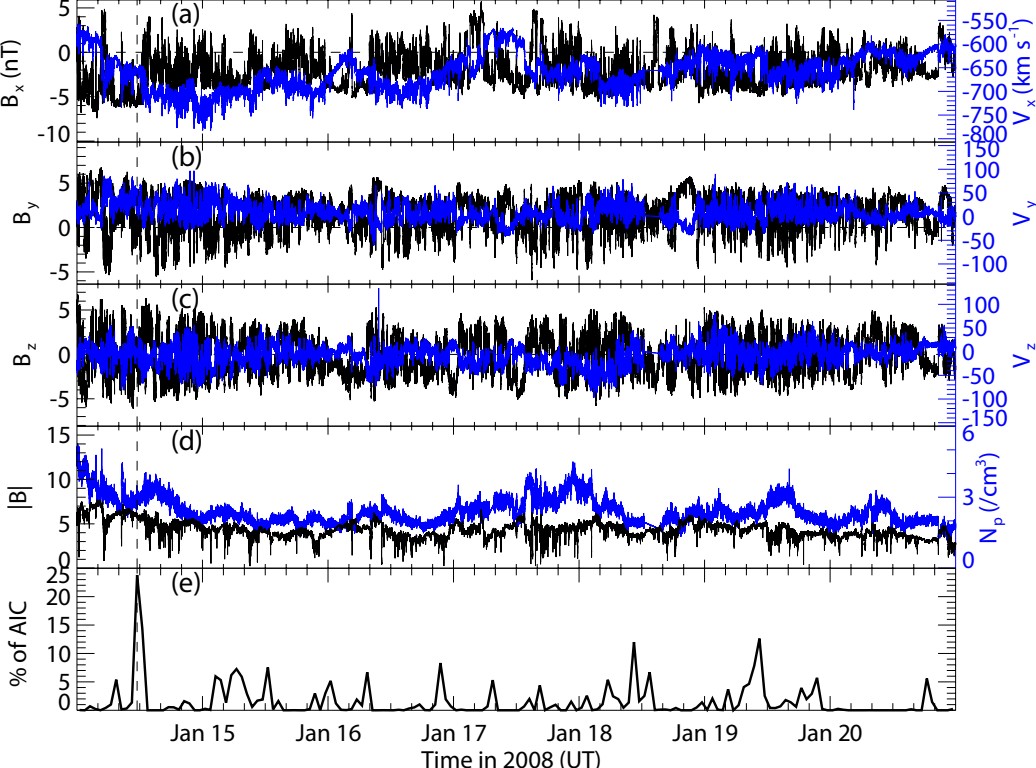

**Figure 1.** (**a–c**) Time series of the three components of both magnetic field vectors (black) and the flow velocity vectors (blue) for the 7-day interval between January 14 and 20 in the year 2008 in GSE coordinates. (**d**) Time series of the magnetic field strength (black) and proton number density (blue). (**e**) Percentage of the data points with $\sigma_m(t_k, \tau_m) < -0.9$ (possible signature of AIC waves) at the scale $\tau_m = 2$ s for each hour of the 7-day interval. The vertical dashed line marks the time instant when a typical case of AIC wave (shown in Figure 3) was found.

Figure 2 shows the trace power spectrum of the magnetic field fluctuations for the 7-day interval in the fast solar wind stream. The black curve corresponds to the spectrum calculated using the Fast Fourier Transform (FFT). The spectrum in the inertial range between 10$^{-3}$ Hz and 0.4 Hz shows nearly Kolmogorov scaling (with spectral index of $f^{-1.65}$), which is consistent with previous observations [5–7]. Then a spectral break occurs at $f_b \sim 0.4$ Hz between the inertial range and the dissipation range. The frequency of the spectral break has been found to be influenced by several factors, such as the solar wind velocity, heliocentric distance, and plasma beta [13,43–46]. Above that break frequency $f_b$, the spectrum steepens and attains an index of $\sim f^{-3.50}$. This index is also comparable to that obtained in previous works [12,47,48]. At frequencies $f > 2$ Hz, the spectrum becomes flatter, possibly due to instrumental noise. In this work, we only focus on the ion kinetic frequency range between 0.52 Hz and 1.21 Hz, in which the spectrum has a power-law

shape and is not contaminated by the noise. This range is marked by the two vertical dashed lines in Figure 2.

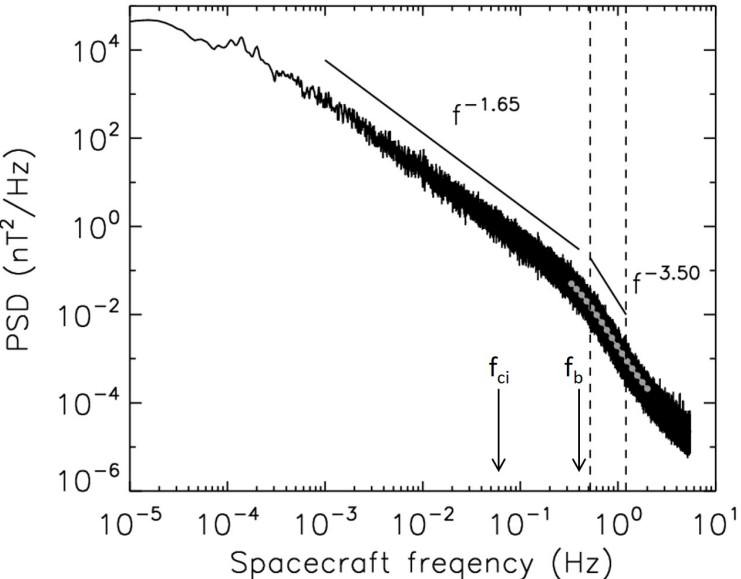

**Figure 2.** Trace power spectral density of the magnetic field as a function of spacecraft frequency as measured in the fast solar wind stream between 14 January 2008 and 20 January 2008 by the *Wind* spacecraft. The black curve corresponds to the power spectrum calculated using the FFT method with 7-point centered smoothing, and the gray dots correspond to the power spectrum computed using the Morlet wavelet method. The spectral indices in the inertial range ($10^{-3}$ Hz–0.4 Hz) and at ion kinetic scales between the two vertical dashed lines (0.52 Hz–1.21 Hz) are also shown. The spectral indices are obtained by applying a least squares fit to the spectra on a log–log plot in the corresponding domains, resulting in the straight lines, respectively. Vertical arrows denote the ion–cyclotron frequency $f_{ci}$ and the break frequency $f_b$.

In order to obtain the magnetic power spectra at different $\theta_{VB}$ angles, we apply the Morlet wavelet method [38] following Horbury et al. [49] and Podesta [50]. First, a discrete wavelet transform is performed on the time series of each magnetic field component $B_i(t_k)$. Here, $i$ denotes the $x$, $y$, and $z$ components in the GSE coordinates, and $t_k = t_0 + k\Delta t$ ($k = 0, 1, \ldots, T/\Delta t - 1$, $\Delta t = 1/11$ s, and $T$ is the length of the data record). The wavelet coefficient of $B_i$ at time $t_k$ and scale $\tau_m$ is calculated from

$$W_i(t_k, \tau_m) = \sum_{n=0}^{T/\Delta t - 1} s_m^{-1/2} \psi^* \left( \frac{t_n - t_k}{s_m} \right) B_i(t_n) \Delta t, \tag{1}$$

where $s_m$ is the wavelet scale ($s_m \approx \frac{\omega_0}{2\pi} \tau_m$, m = 1, 2, ..., 16), $\psi(\eta) = \pi^{-1/4} e^{i\omega_0 \eta} e^{-\eta^2/2}$ is the mother function of the Morlet wavelet, and "*" denotes the complex conjugate. If we set $|W(t_k, \tau_m)|^2 = \sum_{i=x,y,x} |W_i(t_k, \tau_m)|^2$, the trace power spectral density PSD($\tau_m$) can be written as PSD($\tau_m$) $= \frac{4\pi\Delta t}{C\omega_0 T} \sum_{k=0}^{N-1} |W(t_k, \tau_m)|^2$, where $C \approx 1.06$. The gray dots in Figure 2 show the wavelet power spectrum PSD($\tau_m$) for $\tau_m = 0.5$ s–3 s, which coincides with the FFT spectrum very well. The wavelet power spectrum is only presented at $\tau_m = 0.5$ s–3 s here, since this is the domain at which the AIC waves take place, as we will show below. Therefore, we will concentrate on this domain in the following study.

The scale-dependent local mean magnetic field **B**₀ is calculated based on [32,49,50]

$$B_{0i}(t_k, \tau_m) = \sum_{n=0}^{N-1} B_i(t_n) \left| \psi \left( \frac{t_n - t_k}{s_m} \right) \right|^2. \tag{2}$$

The parallel and perpendicular components of the magnetic field for each measurements are then defined with respect to the local $\mathbf{B_0}$. The wavelet coefficients of the parallel component are obtained as follows:

$$W_{\parallel}(t_k, \tau_m) = \frac{W_x(t_k, \tau_m) \cdot B_{0x}(t_k, \tau_m) + W_y(t_k, \tau_m) \cdot B_{0y}(t_k, \tau_m) + W_z(t_k, \tau_m) \cdot B_{0z}(t_k, \tau_m)}{|\mathbf{B_0}(t_k, \tau_m)|}. \tag{3}$$

Accordingly, the wavelet coefficients of the perpendicular components are given by

$$W_{\perp i}(t_k, \tau_m) = W_i(t_k, \tau_m) - \frac{W_{\parallel}(t_k, \tau_m) \cdot B_{0i}(t_k, \tau_m)}{|\mathbf{B_0}(t_k, \tau_m)|}. \tag{4}$$

Then, we obtain the trace power spectral density of the transverse fluctuations from

$$PSD_{\perp}(t_k, \tau_m) = \frac{4\pi\Delta t}{C\omega_0 T} \sum_{i=x,y,x} |W_{\perp i}(t_k, \tau_m)|^2. \tag{5}$$

In the following analysis, we will mainly focus on $PSD_{\perp}$ following Chen et al. [22], since the AIC waves of interest are transverse waves with nearly field-aligned propagation [13,29,32,35,51,52]. Moreover, the perpendicular components of the magnetic fluctuations are dominant, and the ratio between the power of the parallel and perpendicular components could be as small as 0.01–0.4 from both theories and observations [11,22,53,54].

The data of the angle $\theta_{VB}$ between the local $\mathbf{B_0}$ and the solar wind velocity direction are distributed into nine $10°$-wide bins, and then the average power spectrum of the perpendicular components of the magnetic field $PSD_{\perp}(\tau_m)$ in each angular bin is calculated. For example, in the smallest angle bin $0°$–$10°$,

$$PSD_{\perp}(0° - 10°, \tau_m) = \frac{4\pi}{C\omega_0 N(0° - 10°, \tau_m)} \sum_{\substack{k \\ 0° < \theta_{VB}(t_k, \tau_m) < 10°}} PSD_{\perp}(t_k, \tau_m), \tag{6}$$

where $N(0° - 10°, \tau_m)$ is the total number of the data points that satisfy the condition $0° < \theta_{VB}(t_k, \tau_m) < 10°$.

The normalized reduced magnetic helicity $\sigma_m$ can also be obtained from the wavelet coefficients of the perpendicular components of the magnetic fluctuations [32,35,55]

$$\sigma_m(t_k, \tau_m) = \frac{-2Im[W_{\perp y}(t_k, \tau_m) \cdot W_{\perp z}^*(t_k, \tau_m)]}{\sum_{i=x,y,x} |W_{\perp i}(t_k, \tau_m)|^2}. \tag{7}$$

If one wants to study the physical nature of the turbulence at the ion scales, it is necessary to separate the AIC waves, which are coherent events, from the random fluctuations. Here, we consider the wavelet "pixels" with $\sigma_m(t_k, \tau_m) < -0.9$ as the possible signature of left-handed polarized AIC waves, and then study the influence of these waves on the spectral shape of the solar wind turbulence at the ion kinetic scales. The threshold $\sigma_m < -0.9$ is chosen here, since the strong negative helicity corresponds to the data points that are the most likely AIC candidates [32,35,56]. These data points could have the greatest impact on the power spectrum.

In Figure 1e, we present the percentage of the data points with $\sigma_m(t_k, \tau_m) < -0.9$ (possible signature of AIC waves) at the scale $\tau_m = 2$ s for each hour of the 7-day interval. We can see that the highest percentage of the AIC-associated data points (23.5%) appears at the hour between 11:00 UT and 12:00 UT on 14 January 2008. Among some hours, the percentage reaches 2.0–13.0%. In the rest hours (e.g., 16:00 UT–20:00 UT on 15 January 2008, 09:00 UT–15:00 UT on 16 January 2008, and 00:00 UT–17:00 UT on 20 January 2008), the percentage of the AIC-associated data points is close to 0. Therefore, the time periods without AIC waves are dispersively distributed in the fast stream.

Figure 3 gives an example of AIC waves observed between 11:02:49 UT and 11:03:09 UT on 14 January 2008. Figure 3a–c present the time series of the three components of

the magnetic field in GSE coordinates. During this interval, the mean magnetic field is nearly parallel to the radial ($x$) direction, and the two transverse ($y$ and $z$) components of the magnetic field have a sinusoidal-like oscillation with a period of about 2 s. Figure 3d shows the time variation of $\sigma_m(t_k, \tau_m)$ at $\tau_m = 2$ s. We see that the sinusoidal-like oscillation generates a magnetic helicity close to $-1$. In order to better illustrate the oscillation, we plot the motion of the tips of the magnetic fluctuations within the frequency range between 1 s and 3 s in $t - y - z$ coordinates in Figure 3e. Following Lion et al. [37], the fluctuation in the frequency range between 1 s and 3 s is defined as

$$\delta B_i = \bar{B}_{i,1s} - \bar{B}_{i,3s},\tag{8}$$

where $\bar{B}_{i,1s}$ and $\bar{B}_{i,3s}$ are the smoothing average of the magnetic field component $B_i$ ($i = y, z$) with the window width of 1 s and 3 s, respectively. It is clear that the fluctuations show a well-organized circular polarization. Unfortunately, there are no measurements of the proton temperature anisotropy during the plotted interval. In Figure 3f,g, we plot the time-averaged spectra of $PSD(\tau_m)$ and $\sigma_m(\tau_m)$. We can see that a big bump appears around the frequency 0.5 Hz (or around the scale 2 s) in the power spectrum, and within this domain, the value of $\sigma_m$ is close to $-1$.

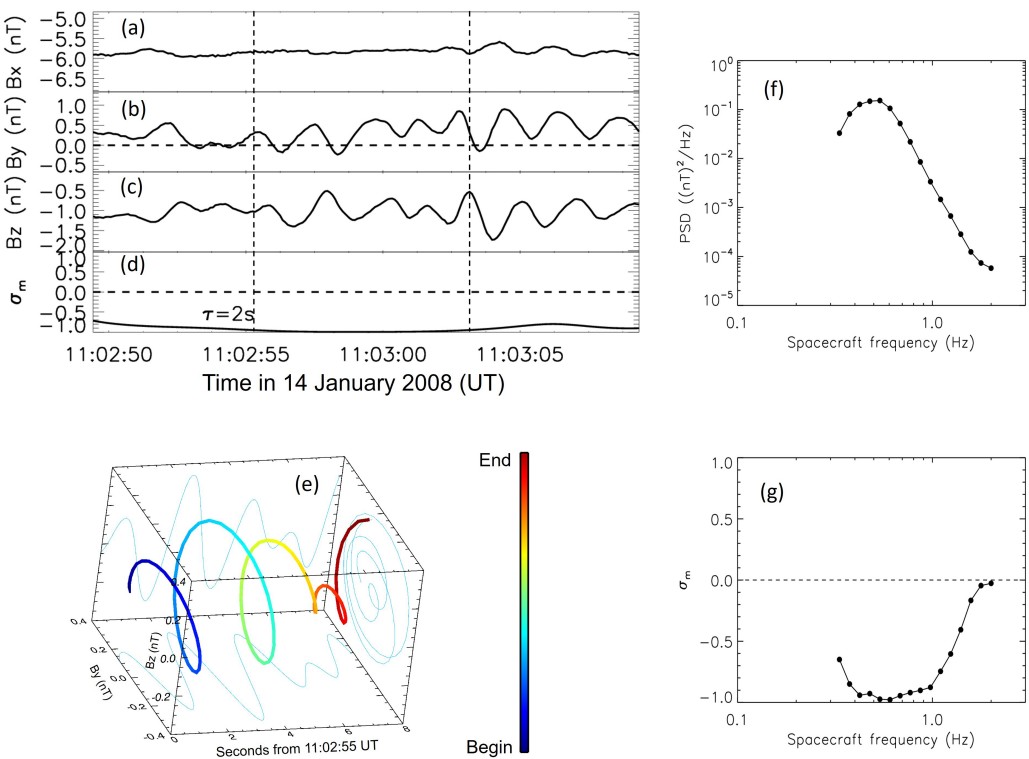

**Figure 3.** An example of AIC waves observed by the *WIND* spacecraft between 11:02:49 UT and 11:03:09 UT on 14 Jaunary 2008. (**a–c**) Time series of the magnetic field components in GSE coordinates. (**d**) Time variation of $\sigma_m(t_k, \tau_m)$ at the scale $\tau_m = 2$ s. (**e**) Motion of the tips of the magnetic field vectors in $t - y - z$ coordinates within the frequency range [1 s, 3 s] between 11:02:55 UT and 11:03:03 UT (denoted by two vertical lines in panels (**a–d**)). (**f,g**) Time-averaged wavelet spectra of PSD and $\sigma_m$.

The wavelet "pixels" with $\sigma_m(t_k, \tau_m) < -0.9$, which distinctly correspond to the AIC waves, are then removed from the original wavelet coefficients, and the average power spectrum $PSD_\perp(\tau_m)$ in each angular bin is recalculated. For example, in the angular bin $0°{-}10°$:

$$PSD'_\perp(0° - 10°, \tau_m) = \frac{4\pi}{C\omega_0 N'(0° - 10°, \tau_m)} \sum_{\substack{k \\ 0° < \theta_{VB}(t_k, \tau_m) < 10° \\ \sigma_m(t_k, \tau_m) > -0.9}} PSD_\perp(t_k, \tau_m), \qquad (9)$$

where $N'(0° - 10°, \tau_m)$ is the total number of the data points that satisfy both $0° < \theta_{VB}(t_k, \tau_m) < 10°$ and $\sigma_m(t_k, \tau_m) > -0.9$. We will compare the spectral indices of the power spectra before and after removing the AIC waves at each $\theta_{VB}$ angular bin in the next section. In order to investigate the dependence of our results on the $\sigma_m$ threshold chosen to identify the AIC waves, we also perform the same analysis as shown in Equation (9) for the $\sigma_m$ thresholds of $-0.8$, $-0.7$, and $-0.6$.

## 3. Results

We first present the number of data points $N(\theta_{VB}, \tau_m)$ used to calculate the average $PSD_\perp(\theta_{VB}, \tau_m)$ at different angular bins and different scales. Based on Equations (6) and (9), we see that the number of data points is a function of both the angular bin and the scale. In the upper panel of Figure 4, we show the number of data points at different $\theta_{VB}$ bins. The upper and lower limits of each error bar denote, respectively, the maximum number and minimum number among all 16 scales of interest at a given $\theta_{VB}$ bin. Different colors correspond to different $\sigma_m$ thresholds when calculating the average $PSD_\perp$, as marked in the panel. We see that the numbers of the data points are always larger than $6 \times 10^4$, which means there are always a large amount of data points that are used to calculate the average $PSD_\perp$. We also note that the number of the data points reaches its maximum value when $30° < \theta_{VB} < 60°$. This can be explained by the effect of the local Parker angle.

In the lower panel of Figure 4, we show the average magnetic power spectra of the transverse fluctuations $PSD_\perp$ for both quasi-parallel (0°–10°) and quasi-perpendicular (80°–90°) angular bins. The symbols and lines in black correspond to the spectra for the original situation before removing the waves. For the spectrum in the parallel angular bin (filled circles), there appears a knee at about 0.5 Hz. This frequency is consistent with the period (2 s) of the AIC wave shown in Figure 3. However, the spectrum in the perpendicular angular bin (unfilled circles) reveals a power-law with no knee, and appears much shallower than the parallel one.

After the AIC waves with $\sigma_m(t_k, \tau_m) < -0.9$ have been removed from the original wavelet coefficients, the remaining power spectra in the parallel angular bin (filled triangles) and perpendicular angular bin (unfilled triangles) are shown in red in the lower panel of Figure 4. The spectrum in the perpendicular angular bin is found to be the same as that in the original data set. This indicates that in the perpendicular direction with 80°–90°, there are nearly no AIC waves with $\sigma_m(t_k, \tau_m) < -0.9$. Yet when comparing the two spectra in the parallel angular bin (black filled circles and red filled triangles), at the frequency range between 0.33 Hz and 1.24 Hz, the new spectrum (red filled triangles) significantly deviates from the original one (black filled circles). At this range, the AIC data points with $\sigma_m(t_k, \tau_m) < -0.9$ account for 1.1% of the whole data set. We find that the knee at ~0.5 Hz has disappeared after the removal of the waves. Therefore, the AIC waves seem to exert substantial influence on the spectral shape of the magnetic field fluctuations in the parallel direction. It is known that the AIC waves mainly propagate parallel to the mean magnetic field. So when the spacecraft samples follow the mean field direction ($\theta_{VB} \sim 0°$), the AIC waves may easily be detected.

Moreover, as shown in Figure 3f, these well-organized AIC waves can produce a power spectrum with a bump around the characteristic period of the waves. Therefore, the spectrum of the original transverse fluctuations in the parallel angular bin around the ion kinetic scales can be divided into two parts: one is the nearly power-law spectrum associated with random turbulent fluctuations (as shown in red filled triangles in the lower panel of Figure 4); the other one is a bump spectrum associated with the AIC waves (similar to that shown in Figure 3f). In the perpendicular angular bin, the spectrum is not associated with the AIC waves, so it does not change significantly during the wave-removal process.

We also show the spectra in the parallel angular bin after removing the data points with $\sigma_m(t_k, \tau_m) < -0.8$ (blue), $\sigma_m(t_k, \tau_m) < -0.7$ (purple), and $\sigma_m(t_k, \tau_m) < -0.6$ (orange) in the lower panel of Figure 4. We see that for the three thresholds, when more data points with relatively strong negative $\sigma_m$ are removed, the spectrum in the parallel angular bin becomes slightly shallower.

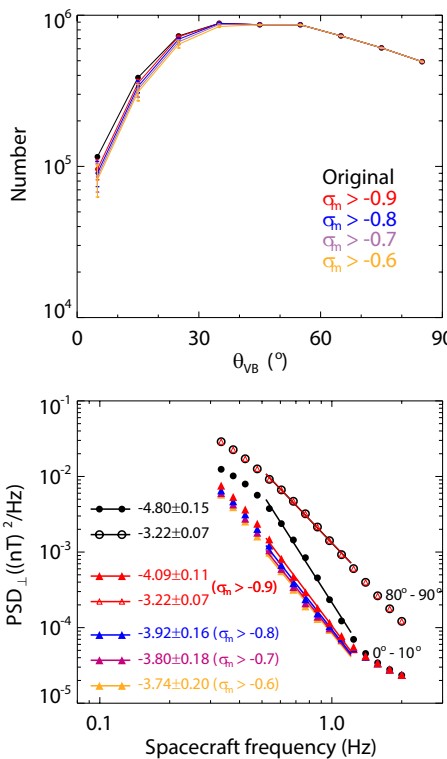

**Figure 4. Upper** panel: Number of data points at different $\theta_{VB}$ bins. At a given $\theta_{VB}$ bin, the upper and lower limits of each error bar denote, respectively, the maximum number and minimum number among all 16 scales of interest. The black dots correspond to the numbers of the data points before the wave-removal process. The dots in red, blue, purple, and orange correspond to different $\sigma_m$ thresholds of $-0.9$, $-0.8$, $-0.7$, and $-0.6$, respectively, when calculating the average $\mathrm{PSD}_\perp$. **Lower** panel: Average power spectra of the transverse fluctuations $\mathrm{PSD}_\perp$ in the parallel ($0°$–$10°$, filled symbols) and perpendicular ($80°$–$90°$, unfilled symbols) angular bins. Different colors correspond to different $\sigma_m$ thresholds as the upper panel. For the perpendicular angular bin, only the original spectrum (black unfilled circles) and the spectrum after removing waves with $\sigma_m(t_k, \tau_m) < -0.9$ (red unfilled triangles) are shown, since the spectra for the other $\sigma_m$ thresholds are nearly the same as the two presented. The spectral indices and their errors, which are obtained from linear fitting of (solid lines) the spectra within the frequency range [0.15, 1.21] Hz on a log–log plot, are also shown.

In Figure 5, we give the spectral index of the transverse fluctuations as a function of the angle $\theta_{VB}$ between $0°$ and $90°$. A minority of data points with $\theta_{VB} > 90°$ are ignored, since the stream of interest is in the fast solar wind associated with an outward magnetic sector. The spectral index is obtained by performing a least-squares fit to the $\mathrm{PSD}_\perp$ spectrum in the frequency range between 0.52 Hz and 1.21 Hz on a log–log plot. The error bar represents the 1-sigma uncertainty estimate of the spectral index based on the least-squares fit. We see that the spectral index of the transverse fluctuations changes gradually from $-4.80 \pm 0.15$ in the parallel angular bin to $-3.22 \pm 0.07$ in the perpendicular angular bin. The trend obtained here of the steepening towards the parallel angular bin is consistent with that shown previously by Chen et al. [22]. But our result seems to be closer to the spectra obtained in the critical balance cascade scenario, which predicts $k^{-7/3}$ in the perpendicular direction and $k^{-5}$ in the parallel direction [23,24].

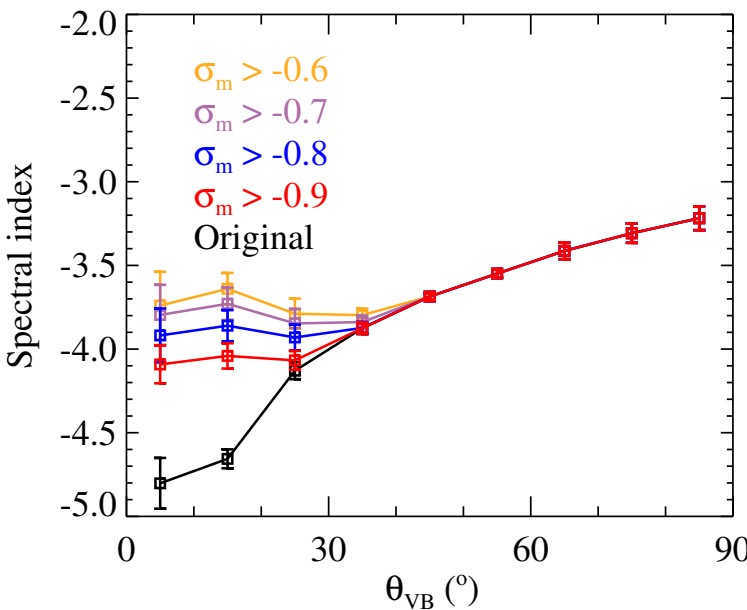

**Figure 5.** Variation of the spectral index of magnetic field within the frequency range [0.52, 1.21] Hz as a function of the $\theta_{VB}$ angle in the original data set (black) and after removal of the AIC waves with $\sigma_m(t_k, \tau_m) < -0.9$ (red), with $\sigma_m(t_k, \tau_m) < -0.8$ (blue), with $\sigma_m(t_k, \tau_m) < -0.7$ (purple), and with $\sigma_m(t_k, \tau_m) < -0.6$ (orange). The spectral index is obtained from performing a least-squares fit to the $PSD_\perp$ spectrum in the frequency range between 0.52 Hz and 1.21 Hz on a log–log plot. The error bar represents the 1-sigma uncertainty estimate of the spectral index based on the least-squares fit.

The colored curves in Figure 5 show the dependence of the newly obtained spectral index on the angle $\theta_{VB}$ after the wave-removal process. It is found that in the smallest parallel angular bin with $0° - 10°$, the slope of the power spectrum becomes $-4.09 \pm 0.11$ when removing the AIC waves with $\sigma_m(t_k, \tau_m) < -0.9$ (red). It is much shallower than that of the original spectrum ($-4.80 \pm 0.15$). This means that after removing the quasi-parallel-propagating AIC waves, the spectrum in the parallel angular bin gets shallower above the break frequency $f_b$. When considering larger angular bins, the difference between the black curve (before removing the waves) and the red curve (after removing the waves with $\sigma_m(t_k, \tau_m) < -0.9$) becomes smaller. And when $\theta_{VB} > 40°$, there seems to be no difference between the curves, probably due to the lack of AIC waves in the quasi-perpendicular direction. Consequently, the anisotropy of the spectral index at the ion kinetic scales becomes weaker.

Besides the results associated with $\sigma_m < -0.9$ (red), we also present the spectral indices and the error bars after removing the data points with $\sigma_m < -0.8$ (blue), $\sigma_m < -0.7$ (purple), and $\sigma_m < -0.6$ (orange) in Figure 5. As mentioned above, the error bars represent the 1-sigma uncertainty estimates of the spectral indices based on the least-squares fits. When considering the error bars, we find that the blue curve (removing $\sigma_m < -0.8$) cannot be well separated from the red one (removing $\sigma_m < -0.9$), even for $\theta_{VB} < 30°$. For example, in the smallest angular bin $0° < \theta_{VB} < 10°$, the spectral index is $\alpha_{ori} = -4.80 \pm 0.15$ originally. When removing the data points with $\sigma_m < -0.9$, the spectral index becomes much larger to $\alpha_{0.9} = -4.09 \pm 0.11$. When removing the data points with $\sigma_m < -0.8$, the spectral index changes only slightly to $\alpha_{0.8} = -3.92 \pm 0.16$. The difference between $\alpha_{ori}$ and $\alpha_{0.9}$ (0.71) is much more significant than the difference between $\alpha_{0.9}$ and $\alpha_{0.8}$ (0.17). The values of $\alpha_{0.7} - \alpha_{0.8} = 0.08$ and $\alpha_{0.6} - \alpha_{0.7} = 0.06$ are even smaller. However, we also note the trend that as the threshold of $\sigma_m$ becomes further away from $-1$, the spectral index at $\theta_{VB} < 30°$ becomes larger gradually. It leads to weaker anisotropy of the spectral index at the ion kinetic scales, since the spectral index in the perpendicular angular bin ($-3.22 \pm 0.07$) nearly does not change. These results suggest that when more data points associated with relatively strong negative helicity are removed, the power spectra at the ion scales get

shallower, especially for $\theta_{VB} < 30°$. It indicates that there still may be small amount of data points-associated AIC waves among the data points with $\sigma_m > -0.9$.

## 4. Discussion and Conclusions

We have studied the influence of the AIC waves on the spectral anisotropy of the magnetic field at ion kinetic scales in a fast solar wind stream by applying the Morelet wavelet method. It has already been determined that the AIC waves can significantly contribute to the observed power spectrum of magnetic fluctuations around ion scales [37]. Here, we find that when we remove the wavelet coefficients with $\sigma_m(t_k, \tau_m) < -0.9$ which correspond to the left-handed polarized AIC waves, the magnetic spectrum of the transverse fluctuations in the parallel angular bin (0°–10°) around the ion kinetic scales (between 0.52 Hz and 1.21 Hz) becomes much shallower and changes from $f^{-4.80\pm0.15}$ to $f^{-4.09\pm0.11}$ (see Figures 4 and 5). When considering larger $\theta_{VB}$ angular bins, we find the difference between the spectral indices of the two situations becomes smaller. And when $\theta_{VB} > 40°$, there seems to be no difference between them, due to the absence of the AIC waves in the quasi-perpendicular direction. In the largest angular bin 80°–90°, the spectral index stays at $-3.22 \pm 0.07$ under the wave-removal process.

The trend of the steepening towards the parallel direction obtained here is consistent with that shown by Chen et al. [22], who reported that the spectral index is $-2.6$ at large angles and $-3$ at small angles by using the structure function method. Since the steepest spectral index that the structure–function technique can determine is only $-3$ [25,26], our work permits us to extend this range, and using the wavelet method, we reliably confirm that the magnetic power spectrum at the ion kinetic scales can be nearly as steep as $k^{-5}$ for the parallel direction.

As shown in Figure 5, the original spectral index in the frequency range between 0.52 Hz and 1.21 Hz changes gradually from $-4.80 \pm 0.15$ in the parallel direction to $-3.22 \pm 0.07$ in the perpendicular direction. After the AIC waves associated with $\sigma_m(t_k, \tau_m) < -0.9$ have been removed from the original data, the anisotropy of the spectral index at the ion kinetic scales becomes weaker ($-4.09 \pm 0.11$ in the parallel direction to $-3.22 \pm 0.07$ in the perpendicular direction). This result indicates that besides the critically balanced cascade [23,24], the spectral anisotropy in this frequency domain could also be partly attributed to the influence of AIC waves. Obviously, more work needs to be conducted to understand the physical nature of the spectral anisotropy around the ion scales in solar wind turbulence. In this work, only one fast solar-wind stream is analyzed. It is necessary to study more intervals to make comparisons and to provide statistical errors in the future.

Moreover, the magnetic helicity $\sigma_m(t_k, \tau_m) < -0.9$ is set as the threshold at which to identify AIC waves, which account for 1.1% of the whole data set at the frequency range [0.33 Hz, 1.24 Hz]. The corresponding wavelet coefficients are removed to study the influence of the AIC waves on the spectral anisotropy. The threshold $\sigma_m < -0.9$ is chosen here, since the strong negative helicity corresponds to the data points that are the most likely AIC candidates. After removing the AIC waves with $\sigma_m < -0.9$, the spectral index of the transverse fluctuations changes from $\alpha_{ori} = -4.80 \pm 0.15$ to $\alpha_{0.9} = -4.09 \pm 0.11$. We also try removing the data points with $\sigma_m < -0.8$, $\sigma_m < -0.7$, and $\sigma_m < -0.6$, and find that the newly obtained spectral index becomes $\alpha_{0.8} = -3.92 \pm 0.16$, $\alpha_{0.7} = -3.80 \pm 0.18$, and $\alpha_{0.6} = -3.74 \pm 0.20$, respectively. This indicates that the data points with $\sigma_m < -0.9$ have greatest impact on the magnetic power spectrum at the ion kinetic scales. It will also be interesting to find some streams without AIC waves, and compare them with the results shown here.

Although the spectrum at the ion kinetic scales becomes shallower during the wave-removal process, it still remains much steeper than the Kolmogorov spectrum ($k^{-5/3}$) in the inertial range. The reason for this steepening is still subject to debate. Some researchers think that it is a consequence of the dissipation of turbulent energy [11,12], while others suggest that it may be related to another turbulent cascade [57,58]. Moreover, as shown by Lion et al. [37], besides the AIC waves, coherent structures also significantly affect the

spectral shape of the magnetic fluctuations. In this work, we have only studied the influence of waves on the magnetic spectra at different angles, but the influence of structures will be considered in the future.

**Author Contributions:** Conceptualization, X.W.; methodology, X.W.; formal analysis, X.W. and L.H.; investigation, X.W., Y.W. and H.Y.; writing—original draft preparation, X.W.; writing—review and editing, X.W.; funding acquisition, X.W. All authors have read and agreed to the published version of the manuscript.

**Funding:** This research has been supported by the National Natural Science Foundation of China (grant nos. 41874199, 41974198, and 41504130), the Fundamental Research Funds for the Central Universities of China (grant nos. KG16152401 and KG16159701), the Btype Strategic Priority Program of the Chinese Academy of Sciences (grant no. XDB41000000), and the pre-research projects on Civil Aerospace Technologies (grant nos. D020103 and D020105) funded by China's National Space Administration (CNSA).

**Data Availability Statement:** WIND data are downloaded from SPDF (http://spdf.gsfc.nasa.gov (accessed on 18 June 2020)). The magnetic field data used in this work are the high-resolution (WI_H2_MFI) data measured via Magnetic Fields Investigation.

**Acknowledgments:** This work at Beihang University is supported by the National Natural Science Foundation of China under contract Nos. 41874199, 41974198, and 41504130. X. Wang is also supported by the Fundamental Research Funds for the Central Universities of China (KG16152401, KG16159701). This work is also supported by the B-type Strategic Priority Program of the Chinese Academy of Sciences (grant No. XDB41000000) and the pre-research projects on Civil Aerospace Technologies No. D020103 and D020105 funded by China's National Space Administration (CNSA). *Wind* data were obtained from SPDF (http://spdf.gsfc.nasa.gov).

**Conflicts of Interest:** The authors declare no conflict of interest.

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
