# Peer review of "Influence of Alfvén Ion–Cyclotron Waves on the Anisotropy of Solar Wind Turbulence at Ion Kinetic Scales"

_universe, doi:10.3390/universe9090399_

Round 1

Reviewer 1 Report

This work addresses the role of Alfven ion cyclotron (AIC) waves in setting the solar wind turbulence spectra at ion scales. The authors did this work by removing AIC wave contributions from the original data and have demonstrated the extent to which AIC waves can make differences in the turbulence spectral index. Overall the paper is well written, the method and results all sound very reasonable to me. I would not hesitate to recommend this paper for publication in this present form. 

Author Response

Thanks very much for the positive evaluation and recommendation. We appreciate your positive opinion and recognition of our work.

Reviewer 2 Report

The manuscript analyses a single long coronal hole wind stream observed with Wind in 2008 which lasted six days. This coronal hole wind stream provides the opportunity to investigate the turbulence spectrum under different aspects. Therefore, time periods with high Alfven-ion-cyclotron (AIC) wave activity are identified and the spectra are compared for different angles between magnetic field and solar wind direction in situations with and without AICs.

The article is well-written and addresses an interesting aspect of the analysis of turbulence. The analysis focuses on kinetic scales in the 0.5-3 s regime.
The analysis is based on a single six-day fast solar wind stream. I agree that a future study should extend this to as many different coronal hole streams as possible.

The power spectra are derived from Morlet wavelets. This is a suitable approach to characterise the fluctuations in the magnetic field. However, whether waves are visible in the wavelet spectrum depends on the coordinate system in which the wavelet analysis is conducted and the current mean magnetic field. Only if the magnetic field happens to be aligned (reasonably well) with one of the coordinate axes along which the Morlet wavelets (or an FFT) are applied, can a wave be visible in this approach. Therefore, the wavelet analysis should not be applied in GSE coordinates but in a local coordinate system with one axis in the direction of the mean magnetic field. I expect that still differences between time periods with and without AIC will be visible (also depending on the magnetic field direction) but the influence of an arbitrary coordinate system should be removed first.

Identifying AICs with the help of the normalized reduced helicity should also be applied in the same magnetic-field-oriented coordinate system. Otherwise, the probability that smaller absolute values of the normalized reduced helicity are "contaminated" with AICs waves will be larger.

I have some additional smaller questions/remark:
- It would also be helpful to show a time series of the magnetic field over this coronal hole wind stream which would help to identify whether this is indeed a single coronal hole wind stream.
- When in the time period are the observations without AICs taken from? Are these (all) already in the rarefaction region?
- Just as a remark: The polarity of the magnetic field should be defined with respect to the local Parker angle. Which angle between solar wind velocity and magnetic field corresponds to  "outward magnetic section" changes considerably with increasing distance from the Sun. In this case, it will not make a difference.
- Figure 4: How well represented are each of the nine angular bins, i.e. does each bin contain a sufficient statistic in the 0.5-3s regime for a well-defined spectrum? What exactly is represented by the error bars here?

minor comments:
page 1, abstract, line 6:
"It is found that around the ion kinetic scales (0.52 Hz − 1.21 Hz), the parallel spectrum has a slope of − 4.59 \pm 0.11 originally."
What "originally" here means becomes clear later in the context of the paper but at this point in the abstract that is not yet obvious.

page 1, section 1, line 18:
"Around the ion kinetic scales, the spectrum is still power-law, but the index becomes much steeper in the dissipation range [8–11]." -> "Around the ion kinetic scales, the spectrum is still a power-law, but the index becomes much steeper in the dissipation range [8–11]."

page 4, section 2, line 130:
"These data points could have greatest impact on the power spectrum." -> "These data points could have the greatest impact on the power spectrum."

page 5:
Figure 2 a) would benefit from being larger.
A color bar for Figure 2 e) is missing.
caption of Figure 2:
"(a-c) Time series of the magnetic field components in the GSE coordinates ." -> "(a-c) Time series of the magnetic field components in GSE coordinates."

page 5, section 2, line 145:
"Unfortunately, there are no measurements about proton temperature anisotropy during the plotted interval." -> "Unfortunately, there are no measurements of the proton temperature anisotropy during the plotted interval."

page 8, section 4, line 218:
"In the largest angular bin 80â—¦ − 90â—¦ , the spectral index stays at − 3.10 ± 0.06 during the wave-removal process." -> "In the largest angular bin 80â—¦ − 90â—¦ , the spectral index stays at − 3.10 ± 0.06 under the wave-removal process."

page 8, section 4, line 234:
"It is necessary to study more intervals to make comparison and to give statistical errors in the future." -> "It is necessary to study more intervals to make comparisons and to give statistical errors in the future."
Good point!

page 8. section 4, line 241:
"We also try removing the data points with $\sigma_m < −0.8, $\sigma_m < −0.7, and $\sigma_m < −0.6, and find that the newly obtained anisotropy is not significantly different from that of $\sigma_m < −0.9 as shown in Figure 4 in red."
Have these differences been tested for statistical significance? If not, how were the different?

page 8, section 4, line 245:
"It will also be interesting to find some streams without AIC waves, and compare with the result shown here." -> "It will also be interesting to find some streams without AIC waves, and compare with the results shown here."

As far as I can evaluate that the English quality is good but can be improved further.

Reviewer 3 Report

The article "Influence of Alfvén-ion-cyclotron waves on the anisotropy of solar wind turbulence at ion kinetic scales"  is showing how the Alfven ion cyclotron waves influence the spectra in the parallel angular bins and not in the perpendicular angular bins. The work done suits the journal and could be acceptable for publication after some important revisions. There are some major issues with the usage of terminology and their comparison with previous works. 

1.) Authors are confused between parallel and perpendicular spectra to spectra corresponding to the parallel angular bins and perpendicular angular bins,(eg: lines 7, 9 and in many places) I would like the authors to rectify that. Instead of parallel/perpendicular spectrum, authors should clearly mention spectrum in the parallel/perpendicular angular bin.  The difference can be understood from Figure 4 of Chen et al 2010 "Anisotropy of Solar Wind Turbulence between Ion and Electron Scales"  where authors show the parallel and perpendicular fluctuations/spectrum.

2.) Lines 128-130 " The threshold σm < −0.9 is 128 chosen here, since the strong negative helicity corresponds to the data points that are the 129 most likely AIC candidates."

Citations are required.

3)Lines 170- 172 "But our result seems to be closer to the spectra obtained in the critical balance cascade scenario, which predicts k−7/3 for the perpendicular spectrum and k−5 for 171 the parallel spectrum [24,25]. 

As mentioned in comment 1, your results are not showing any parallel or perpendicular spectrum, you are showing the whole spectrum separated by angular bins, therefore you might not be able to compare your observations with the above statement.

4.) Lines 173-176 have to be used before describing Figure 4 as they are part of Figure 3 description

5.) 218-220 "The trend of the steepening towards the parallel direction obtained here is consistent with that shown by Chen et al. [23], who reported that the spectral index is −2.6 at large angles and −3 at small angles by using the structure function method"

Should mention that the observations of Chen et al you suggest are for transverse fluctuations.

I suggest that if the authors want to talk and compare their results to that of Chen et al 2010, they should be performing the same analysis for the fluctuations/PSD in the parallel and perpendicular direction as shown in Figure 4 of Chen et al 2010, https://arxiv.org/pdf/1002.2539.pdf. Then they can mention parallel and perpendicular spectrum. 

Round 2

Reviewer 2 Report

I am satisfied with the revision of the paper. All my concerns have been addressed adequately. Thus, from my point of view, the manuscript is now ready for publication.

Two (very minor) comments/suggestions:
page 7, line 192:
"From Equations (6) and (9), we see that the number is the function of both the angular bin and the scale." -> "From Equations (6) and (9), we see that the number of data points is a function of both the angular bin and the scale."

Figure 4:
I suggest to change the color scheme. Yellow is difficult to read and the current combination of green and red, and blue and yellow is not very color-blind friendly.

Reviewer 3 Report

After going through the replies and changes made to improve the clarity of the work, I feel the article is in good shape for publication.

Author Response

Thanks very much for the recommendation.